# Causal Forecasting:
# Generalization Bounds for Autoregressive Models

**Leena Chennuru Vankadara**[*1]   **Philipp Michael Faller**[2]   **Michaela Hardt**[2]   **Lenon Minorics**[2]

**Debarghya Ghoshdastidar**[3]                          **Dominik Janzing**[2]

[1]University of Tübingen
[2]Amazon Research
[3]Technical University of Munich, Munich Data Science Institute.

## Abstract

Despite the increasing relevance of forecasting methods, causal implications of these algorithms remain largely unexplored. This is concerning considering that, even under simplifying assumptions such as causal sufficiency, the statistical risk of a model can differ significantly from its *causal risk*. Here, we study the problem of *causal generalization*—generalizing from the observational to interventional distributions—in forecasting. Our goal is to find answers to the question: How does the efficacy of an autoregressive (VAR) model in predicting statistical associations compare with its ability to predict under interventions? To this end, we introduce the framework of *causal learning theory* for forecasting. Using this framework, we obtain a characterization of the difference between statistical and causal risks, which helps identify sources of divergence between them. Under causal sufficiency, the problem of causal generalization amounts to learning under covariate shifts albeit with additional structure (restriction to interventional distributions under the VAR model). This structure allows us to obtain uniform convergence bounds on causal generalizability for the class of VAR models. To the best of our knowledge, this is the first work that provides theoretical guarantees for causal generalization in the time-series setting.

## 1 INTRODUCTION

Forecasting algorithms are increasingly relevant in a variety of applications including meteorology, climatology, economics, and business. While traditional economic modelling relies on relatively simple time series models (Brockwell et al. 1991), e.g., autoregressive models, or methods like co-integration, modern business planning heavily uses neural networks for forecasting (Faloutsos et al. 2018; Januschowski et al. 2020; Salinas et al. 2020). Despite the advancements of forecast quality, causal implications are not yet well understood. There has been notable progress in 'explainable' models in the sense of feature relevance (Lundberg et al. 2017; Molnar 2019; Janzing et al. 2020; Wang et al. 2020) with potential applications in forecasting. Furthermore, specialized models (Hatt et al. 2021; Bica et al. 2020; Lim et al. 2018) have shown remarkable success for causal inference in forecasting.

It is common practice in business and econometrics to learn statistical forecasting models and interpret them causally. In practice, while forecasting models tend to agree on their statistical predictions, they can differ substantially on their causal predictions (see Figure 1 for an example). In particular, this practice is considered justified under simplifying assumptions such as causal sufficiency and the absence of contemporaneous effects (see for instance Hyvärinen et al. (2010, Section 1)). Here, we are interested in the fundamental question: what is the relation between the statistical predictability of a forecasting model and its causal generalizability — ability to predict under interventions.

We argue that even for very simple models and even under simplifying assumptions such as causal sufficiency and absence of contemporaneous influence, causal interpretation of forecasting models is non-trivial. To appreciate the challenges, consider a simple example of a process with strongly correlated observations where $x_t \approx x_{t-1}$, and hence $x_t \approx x_{t-2}$. These observations can be explained either by a causal model with a strong influence of $x_{t-1}$ on $x_t$ or a causal model with a strong influence from $x_{t-2}$ on $x_t$. The difference between the models gets apparent when an intervention randomizes $x_{t-1}$ and $x_{t-2}$ independently. Then, predictions become hard, particularly when $x_{t-1}$ and $x_{t-2}$ are set to significantly different values. While both models

---

*Part of this work was completed while the author was at Amazon Research.

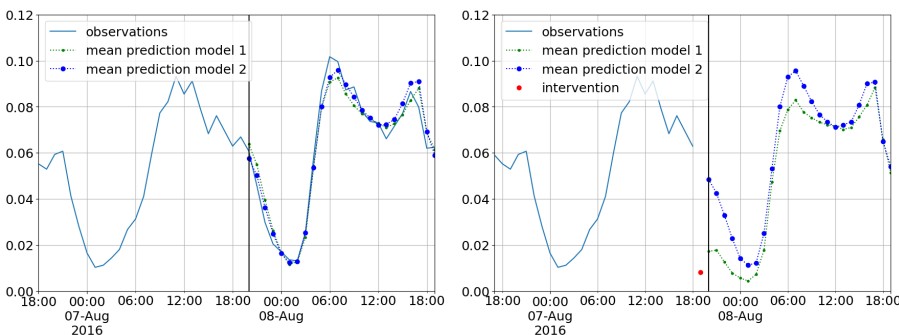

Figure 1: An example time series with predictions of two DeepAR models (top) under an intervention in red (bottom) on the Traffic dataset. While we do not know the ground-truth, we see that two models disagree when faced with an intervention more than on the in-distribution forecasting. Since at most one of them can be right, we conclude that at least the other one makes a notable forecasting error under the intervention.

are similar in their statistical predictions, they differ substantially in their *causal predictions*. This example already shows that, even in a simple setting, causal and statistical predictability can differ significantly. The question of causal generalization is thus practically relevant and non-trivial and begs for a better theoretical understanding.

Specifically, we consider the simple class of vector autoregressive models (VAR) and ask the question

*How does the efficacy of an autoregressive model in predicting statistical associations compare with its ability to predict under interventions?*

These models are widely applied in domains ranging from econometrics (Lütkepohl 2009; Grabowski et al. 2020) and finance (Zivot et al. 2006) to neuroscience (Valdés-Sosa et al. 2005).

**Connection to Covariate Shift.** The problem of causal generalization is closely related to the problem of covariate shift. To see this, we first ignore the time series setting and consider the scenario where a variable $Y$ should be predicted from a variable $X$, which is known not to be an effect of $Y$. If there is no common cause of $X$ and $Y$, that is, we assume causal sufficiency (Spirtes et al. 1993), the statistical relation between $X$ and $Y$ is entirely due to the influence of $X$ on $Y$. Therefore, the observational and interventional conditionals coincide ($P_{Y|x=x^*} = P_{Y|do(x=x^*)}$ in Pearl's language (Pearl 2009)) and the true parameters would be optimal both from a statistical and causal perspective. However, due to *estimation bias*, a prediction model learned using finite samples from $P_x$ may perform poorly when randomized interventions draw $x$-values from a different distribution $\tilde{P}_X$, which is the usual covariate shift scenario (Sugiyama et al. 2012). In our setting, $X$ and $Y$ are represented by the past and the present values of a (possibly multivariate) time series, respectively. Accordingly, we focus on interventional distributions that are natural for this setting: independent interventions at different time points and components of the

multivariate process. Hence, we have additional structure in comparison with the standard covariate shift problem. We are not aware of any theoretical work on covariate shift in the time-series setting. Nevertheless, we describe the connections to learning theory in the standard covariate shift setting and other related work in Section 6.

**Our Contributions.** Our central goal in this work is to develop a formal and thorough understanding of causal generalization for the class of VAR models.

a. To this end, we introduce a framework of causal learning theory for forecasting to analyze when forecasting models can generalize from the *observational* to the *interventional distributions* (Section 2). This is closely related to the setting of learning under domain adaptation.

b. Using this framework, we provide a characterization of the difference in the statistical and *causal* risks (Section 3). Such a characterization allows us to identify the sources of divergence between the two quantities. Our results show that the strength of correlation of the underlying process plays a key role in determining causal generalizability. They also highlight that already for simple models, causal and statistical errors can even diverge.

c. Further, we provide finite-sample, uniform convergence bounds on causal generalization for the class of VAR models (Section 3). Our simulations demonstrate that our bounds indeed capture the key drivers of causal generalization. To the best of our knowledge, this is the first work that provides theoretical guarantees for causal generalization of any kind in the time-series setting.

d. As a by-product of our analysis, we provide an explicit characterization of the powers of a companion matrix (see Section 2) using symmetric Schur polynomials (Macdonald 1998) of its eigenvalues (Lemma 2) which, to the best of our knowledge, has not been noted in the literature. This result could be of independent interest in theoretical endeavors that build upon companion matrices which, for instance, are ubiquitous in stochastic processes and in

Linear-Time-Invariant dynamical systems (Davison 1976; Melnyk et al. 2016).

e. We conduct experiments with a variety of deep neural networks on real data. Our experiments approach causal risks in this setting and explore its relationship to uncertainty.

# 2 CAUSAL LEARNING THEORY FOR FORECASTING

In this section, we introduce a framework to formally evaluate the quality of a forecasting model with respect to prediction and the validity of its causal implications. We refer to this framework as causal learning theory for forecasting. First, we introduce some relevant notation.

**Notation.** For any stochastic process $\{x_t\}_{t \in \mathbb{Z}} \in \mathbb{R}^d$, we use $\mathbf{x}_{t-\omega}^n = \{x_{t-\omega-n+1}, \cdots, x_{t-\omega-1}, x_{t-\omega}\}$ to denote the *set* of $x_{t-\omega}$ and the $n-1$ variables in the past of $x_{t-\omega}$. We distinguish this from $y_t^n$ which denotes the *vector* $(x_t, x_{t-1}, \cdots, x_{t-n+1})^T \in \mathbb{R}^{nd}$. When it is clear from context, to reduce cumbersome notation, we simply use $y_t$. For any random variable $x$, $\mathbb{E}[x]$ denotes its expectation. For any matrix $A$, we use $A_{i:}$ and $A_{:j}$ to denote the $i$th row and $j$th column of $A$ respectively. We use $A_{1k}^j$ to denote the $(1, k)$th element of $A^j$. For any vector $x_t$ at time $t$, we use $x_{t,i}$ to denote the $i$th element of $x_t$. We use $\lambda_{\max}(A), \lambda_{\min}(A), \kappa(A) = \lambda_{\max}(A)/\lambda_{\min}(A)$ to denote the maximum and minimum eigenvalues and the condition number of $A$ respectively. $\mathbb{I}_p$ denotes the identity matrix of size $p$, $\mathbb{N}, \mathbb{Z}$ denote the set of natural numbers and integers respectively and $[n]$ denotes the set $\{1, 2, \cdots n\}$.

To evaluate the statistical and causal efficacy of an estimator we introduce the notions of statistical and *causal* forecast risks. To define statistical forecast risk, we consider the setting of $\omega-$step forecasting where the goal is to predict $x_t$ from observations $\mathbf{x}_{t-\omega}^n$ drawn from a stochastic process $\{x_t\}_{t \in \mathbb{Z}}$ for some $\omega \in \mathbb{N}$. To define the causal forecast risk, we consider interventions on $x_{t-\omega,i}$ for some $i \in [d]$.[1]

**Definition 2.1 (Statistical forecast error).** The statistical forecast error of an estimator $\hat{f}$ in the prediction of a target variable $x_t$ from $\mathbf{x}_{t-\omega}^n$, drawn from the *observational distribution*, can be defined as

$$\mathcal{S}_\omega = \mathbb{E}_{\mathbb{P}(x_t, \mathbf{x}_{t-\omega}^n)}\big[(x_t - \hat{f}(x_{t-\omega}^n))^2\big]. \qquad (1)$$

The empirical counterpart ($\hat{\mathcal{S}}_\omega$), is defined naturally by replacing the expectation by the empirical mean.

For causal questions, we want to investigate the behavior of a model under interventions. Here, we consider atomic

---

[1]The results for simultaneous interventions are qualitatively similar to those of interventions on single variables, and for ease of exposition, we present our discussion in the latter case.

interventions. Using Pearl's do notation (Pearl 2009), an atomic intervention $do(x = x^*)$ refers to *setting* the variable $x$ to some value $x^*$.

**Definition 2.2 (Causal errors).** The interventional forecast error of $\hat{f}$ in predicting the *effect of an intervention* $do(x_{t-\omega,i} = x_{t-\omega,i}^*)$, on target variable $x_t$ is defined as

$$\mathcal{G}_{do_{\omega,i}} = \mathbb{E}_{\mathbb{P}_{do_{\omega,i}}(x_t, \mathbf{x}_{t-\omega}^n)}\big[(x_t - \hat{f}(x_{t-\omega}^n))^2\big], \qquad (2)$$

where $do_{\omega,i}$ is shorthand for $do(x_{t-\omega,i} = x_{t-\omega,i}^*)$ and $\mathbb{P}_{do_{\omega,i}}$ denotes the distribution induced by the intervention $do(x_{t-\omega,i} = x_{t-\omega,i}^*)$. To isolate from the dependence on specific values that the intervened variables are set to, we present our results via the notion of *average causal error*. It is defined as the expected interventional error for interventions drawn from the marginal distribution of $x_{t-\omega,i}$ since it provides a natural scale at which the statistical and causal errors can be compared.

$$\mathcal{G}_{\omega,i} = \mathbb{E}_{x_{t-\omega,i}^* \sim \mathbb{P}(x_{t-\omega,i})}\big[\mathcal{G}_{do_{\omega,i}}\big]. \qquad (3)$$

**Statistical and Causal Learning Theory.** Consider the standard framework of statistical learning in time-series prediction. For any stochastic process $\{x_t\}_{t \in \mathbb{Z}}$ taking values in $\mathcal{X}$, given a loss function $l : \mathcal{X} \times \mathcal{X} \to \mathbb{R}^+$, the goal of statistical learning is to learn a function $f_{\mathcal{S}}^*$ that achieves the optimal statistical risk $\mathcal{S}^\omega(f)$:

Since the true process is unknown, the empirical average ($\hat{\mathcal{S}}^\omega$) of generalization risk is used to estimate $\mathcal{S}^\omega$. Statistical generalization bounds of the form: $\mathcal{S}^\omega(f) < \hat{\mathcal{S}}^\omega(f) + \mathcal{C}(\mathcal{F}, n)$ are then used to provide guarantees on the uniform deviation of empirical risk from expected risk given sufficiently many samples and when the "complexity" of the function class is small.

Analogously, the goal of *causal learning* is to find a function $f_{\mathcal{G}}^*$ that achieves the optimal *causal* risk $\mathcal{G}^\omega(f)$

In contrast to statistical learning, the empirical averages of the causal error cannot be utilized to estimate $\mathcal{G}_\omega$ since we often do not have access to data from the interventional distributions. Instead, we are only provided with data from the observational/statistical distribution of the stochastic process and the goal of causal learning theory is to understand, to what extent is it possible to provide *causal generalization* guarantees of the form: $\mathcal{G}^\omega(f) < \hat{\mathcal{S}}^\omega(f) + \mathcal{C}(\mathcal{F}, n)$.

To summarize, we ask: Can the predictors in $\mathcal{F}$ generalize from the *empirical observational distribution* to the *true interventional distribution* assuming that we control the complexity of $\mathcal{F}$ and that we observe sufficiently many samples drawn from the observational distribution? One cannot address this question in a very general setting and would need model assumptions to make any meaningful statements. To this end, we now formally introduce our problem setup and some preliminaries. We provide additional relevant background in the Appendix 1.

**Statistical and Causal Models.** We assume that the stochastic process $\{x_t\}_{t\in\mathbb{Z}} \in \mathbb{R}^d$ follows a weakly stationary vector autoregressive model(VAR(p)) of order $p$ for some $p, d \in \mathbb{N}$ which is defined as

$$x_t = A_1 x_{t-1} + A_2 x_{t-2} + \cdots A_P x_{t-p} + \epsilon_t, \quad (4)$$

where $x_t \in \mathbb{R}^d$ is a vector-valued time-series, for all $i \in [p]$, $A_i \in \mathbb{R}^{d\times d}$ are the coefficients of the VAR model, and $\epsilon_t \in \mathbb{R}^d$ denotes the noise vector such that $\mathbb{E}[\epsilon_t] = 0$ and $\mathbb{E}[\epsilon_t \epsilon_{t+h}^T] = \Sigma_\epsilon$ if $h = 0$ and 0 otherwise. For some $\sigma_\epsilon^2 > 0$, we simply set $\Sigma_\epsilon = \sigma_\epsilon^2 \mathbb{I}$ for enhanced readability. Our results can be easily generalized to arbitrary covariance matrices by means of the spectral properties $(\lambda_{\min}, \lambda_{\max})$ of $\Sigma_\epsilon$. The autocovariance matrix of $\{x_t\}_{t\in\mathbb{Z}}$ plays a central role in our results and analysis. For any $n \in \mathbb{N}$, we use $\Sigma_n$ to denote the autocovariance matrix of size $n$ defined as $\mathbb{E}[(y_t^n - \mathbb{E}[y_t^n])(y_t^n - \mathbb{E}[y_t^n])^T]$. It is convenient to rewrite a VAR model of order $p$ in Equation (4) as a VAR(1) model, $y_t = A y_{t-1} + e_t$, where $y_t \in \mathbb{R}^{dp}, e_t \in \mathbb{R}^{dp}$ are defined as $y_t = (x_t, x_{t-i}, \cdots, x_{t-p+1})^T$, $e_t = (\epsilon_t, 0, \cdots, 0)^T$, and $A \in \mathbb{R}^{dp\times dp}$ is a *(multi) companion matrix* defined as:

$$A = \begin{pmatrix} A_1 & A_2 & \cdots & A_{p-1} & A_p \\ I & 0 & \cdots & 0 & 0 \\ 0 & I & \cdots & 0 & 0 \\ \vdots & \vdots & \cdots & \vdots & \vdots \\ 0 & 0 & \cdots & I & 0 \end{pmatrix}. \quad (5)$$

The eigenvalues of the multi-companion matrix $A$ fully characterize the stability and stationarity of the VAR process. For a VAR(p) process to be weakly stationary, that is for the mean and the covariance of the process to not change over time, the eigenvalues of $A$, which satisfy

$$\det|\mathbb{I}_d \lambda^p - A_1 \lambda^{p-1} - A_2 \lambda^{p-2} - \cdots - A_p| = 0, \quad (6)$$

are constrained to not lie on the unit circle. If the magnitude of all the eigenvalues are $|\lambda_i| < 1$, then the process is stable, that is, its values do not diverge (Lütkepohl 2013).

**Causal Models.** Under the assumptions of causal sufficiency and absence of contemporaneous influences, a causal interpretation of the VAR model in (4) as structural equations naturally yields the corresponding causal model. We consider the family of all VAR models as our function class $\mathcal{F}$ of statistical and causal estimators.

## 3 CAUSAL GENERALIZATION FOR VAR

In this section, we present causal generalization bounds for the family of VAR models under atomic interventions. We first provide an overview of our results in the more general case of VAR(p) models and later provide a thorough interpretation of the results, often by deriving simplified versions of the results for AR(p) models. We begin by providing

an exact characterization of the difference in statistical and causal errors in terms of the model and estimated parameters and the autocovariance matrix of the underlying process.

**Lemma 1 (Difference in Causal and Statistical errors (VAR)).** *Consider a vector-valued time series $\{x_t\}_{t\in\mathbb{Z}} \in \mathbb{R}^d$, following a VAR(q) process parameterized by $\{A_1, A_2, \cdots A_q\}$. Let $\nu = \max\{p, q\}$. For any VAR(p) model $f$ with parameters $\{\hat{A}_1, \hat{A}_2, \cdots \hat{A}_p\}$,*

$$|\mathcal{G}_{\omega,i} - \mathcal{S}_\omega| = 2\left|(A_{ii}^\omega - \hat{A}_{ii}^\omega)\sum_{k\neq i}^{d\nu}(A_{ik}^\omega - \hat{A}_{ik}^\omega)\Sigma_{ik}^\nu\right|,$$

*where $\Sigma^\nu$ denotes the autocovariance matrix of $x_t$ of size $\nu$, $A$ is a multi-companion matrix of the form described in (5) with the first $d$ rows populated by $\{A_1', A_2', \cdots A_\nu'\}$, with $A_l'$ defined as $A_l$ for all $l \leq p$ and as $\mathbf{0}_{d\times d}$ for all $l > p$. $\hat{A}$ is analogously defined.*

Building on Lemma 1, we establish that the condition number of the autocovariance matrix of the underlying process controls causal generalizability from the *observational* to *interventional* distributions.

**Proposition 1 (Stability Controls Causal Generalization (VAR)).** *Let $\{x_t\}_{t\in\mathbb{Z}}$ follow a VAR(q) process for some $q \in \mathbb{N}$. For any VAR(p) model,*

$$|\mathcal{G}_{\omega,i} - \mathcal{S}_\omega| \leq (2\kappa(\Sigma^\nu) - 1)(\mathcal{S}_\omega - \sigma_\epsilon^2), \quad (7)$$

*where $\kappa(\Sigma^\nu)$ denotes the condition number of the autocovariance matrix $\Sigma^\nu$. Further, one can construct processes where equality holds upto a small constant factor.*

The result states that the difference in expected causal and statistical errors is controlled by the *condition number* of the autocovaraince matrix of size $\max\{p, q\}$. It also states that without incorporating additional information, one cannot obtain a much tighter bound which is also verified by our experiments in Section 4. The condition number of the autocovariance matrix can get arbitrarily large as the process gets closer to the boundary of the stability domain. This result therefore shows that even for very simple classes of forecasting models, causal interpretations can get challenging. We later provide a detailed interpretation of this result and provide an explicit bound on $\kappa(\Sigma_\nu)$ in terms of the stability parameter for AR(p) models (Corollary 2).

Proposition 1 allows us to employ generalization bounds for time-series (Yu 1994; Meir 2000; Mohri et al. 2009; McDonald et al. 2017) to derive finite-sample *causal generalization bounds* for VAR models. In particular, we utilize Rademacher complexity bounds for generalization in time-series under mixing conditions (Mohri et al. 2009) to derive Theorem 1.

**Theorem 1 (Finite sample bounds for VAR(p) models).**
*Let $\mathcal{F}$ denote the family of all VAR models of dimension $d$ and order $p$. For any $n > \max\{p, q\} \in \mathbb{N}$, let $\mu, m > 0$ be integers such that $2\mu m = n$ and $\delta > 2(\mu-1)\rho^m$ for a fixed constant $0 < \rho < 1$ determined by the underlying process. Let $\{x_1, x_2, \cdots x_n\} \in \mathbb{R}^d$ be a finite sample drawn from a VAR(q) process. Then, simultaneously for every $f \in \mathcal{F}$, under the square loss truncated at $M$, with probability at least $1 - \delta$,*

$$\mathcal{G}_{\omega,i} \leq \zeta\hat{\mathcal{S}}_\omega + \zeta\widehat{\mathfrak{R}}_\mu(\mathcal{F}) + 3\zeta M\sqrt{\frac{\log\frac{4}{\delta'}}{2\mu}} \quad (8)$$

*where $\zeta = 2\kappa(\Sigma^\nu)$, $\delta' = \delta - 2(\mu-1)\rho^m$, and $\widehat{\mathfrak{R}}_\mu(\mathcal{F})$ denotes the empirical Rademacher complexity of $\mathcal{F}$.*

Our causal generalization bound in Theorem 1 suggests that, given sufficiently many samples, the true causal error can be guaranteed to be close to empirical statistical error if our VAR models come from a class with a small Rademacher complexity, particularly when the process is associated with a small stability parameter.

We now focus on providing a detailed interpretation of our results. First, we take a minor detour to present a technical result (Lemma 2) which is useful both in deriving some of our main results as well as in interpreting them.

**Lemma 2 (Expressing powers of a companion matrix using symmetric polynomials).** *For a companion matrix $A$ with distinct eigenvalues, for any $k \in [p]$, the $(1, k)$th element of $A^j$, can be expressed using Schur polynomials of the eigenvalues $\lambda = \{\lambda_1, \lambda_2, \cdots \lambda_p\}$ of $A$, that is, $A^j_{1,k} = S_{j,k}(\lambda)$, where $S_{j,k}(\lambda)$ refers to the Schur polynomial indexed by $K = \{j, 1, \cdots k - 1 \text{ times} \cdots, 1, 0, \cdots, 0\}$.*

Lemma 2 shows that the coefficients of the powers of a companion matrix can be fully characterized using symmetric Schur polynomials of its eigenvalues. A good overview of these polynomials can be found in Chaugule et al. (2019). An advantage of expressing the coefficients using symmetric Schur polynomials is that these polynomials have been a subject of extensive research in combinatorics and an equivalence between several alternate definitions has been established. To name a few, Cauchy's bialternant expression, (Cauchy 1815; Jacobi 1841), the combinatorial formula (Macdonald 1998) or Jacobi–Trudi identity (Jacobi 1841) are all equivalent ways to define Schur polynomials. It is therefore possible and often beneficial to choose the definition that yields the most useful notion for the context. We utilize this connection to interpret our results. First, for easier interpretation, we simplify Lemma 1 to the following result for scalar AR models.

**Corollary 1 (Difference in Causal and Statistical errors (AR)).** *Let $\{x_t\}_{t \in \mathbb{Z}}$ follow an AR(q) process. Then, for any*

*AR(p) model with parameters $\hat{A}$,*

$$|\mathcal{G}_{do_\omega} - \mathcal{S}_\omega| = 2\left|(A^\omega_{11} - \hat{A}^\omega_{11})\sum_{k=2}^\nu (A^\omega_{1k} - \hat{A}^\omega_{1k})\gamma_{k-1}\right|, \quad (9)$$

*where, for any $k \in \mathbb{N}$, $\gamma_k$ denotes the autocovariance of $\{x_t\}_{t \in \mathbb{Z}}$ with lag $k$. $A$ and $\hat{A}$ are the corresponding companion matrices of the model and estimated parameters as defined in Lemma 1.*

Lemma 1 identifies factors that control causal generalizability. We now describe them.

**Correlations control causal generalizability.** Recall our motivating example of the two highly correlated time-series where the casual and statistical errors diverge. Intuitively, one would therefore expect that large correlations among time series potentially induce large differences between observational and interventional distributions. The quantitative dependence of causal generalizability on the correlation structure of the process is, however, less obvious. Lemma 1 confirms the intuition and shows that correlations between the intervened time-series $x_{t-\omega,i}$ across both the components and time instances in $\mathbf{x}_{t-\omega}$ control generalizability from observational to the interventional distributions.

**High-dimensional and higher-order processes can hurt generalization.** For high-dimensional processes it is not unlikely to have strong correlations across components, which may obscure causal relations in the same way as strong correlations across time does for univariate processes. Lemma 1 also supports this intuition and shows that strong correlations across components as well as time instances play a role. With increasing order or dimension of the processes, larger orders of covariances across time and dimensions could entail poor causal generalizability.

**Dependence on $\omega$.** The dependence of the error on $\omega$ arises through the elements of the matrix power $A^k$. A simple computation shows that, even for an AR(2) model, the dependence of these coefficients on the model parameters is asymmetric and highly intricate. However, using the Cauchy's bialternant formulation of Schur polynomials, we have that for any AR(p) model, the coefficients $A^\omega_{1k}$ can be expressed as $A^\omega_{1k} = (-1)^{k+1}\frac{\sum_{i=1}^p \lambda_i^{p+\omega-1}e_k(\lambda_i)}{\det\left|\{\lambda_k^{p-k'}\}_{k,k'\in[p]}\right|}$, where $e_k(\lambda_i)$ refers to the elementary symmetric polynomial of order $k$ and with variables $\{\lambda_1, \cdots \lambda_{i-1}, \lambda_{i+1}, \cdots, \lambda_p\}$. While this is not the most interpretable definition per se, the dependence of the coefficients on $\omega$ is easily understood and it is easy to verify that if the underlying model as well as the estimated model are both stable ($|\lambda| < 1$), the coefficients and hence the difference in errors exponentially decays with interventions arbitrarily in the past of the target variable and if either of the process is not stable ($|\lambda| > 1$), the difference can indeed diverge.

Proposition 1 allows us to obtain a high-level perspective on causal generalizability. It states that the condition number of the autocovariance matrix controls causal generalizability. Both the maximum and the minimum eigenvalue of the autocovariance matrix (and hence the condition number) can be used as a measure of stability and hence determine the strength of correlation of the underlying process (Basu et al. 2015; Melnyk et al. 2016). As the process gets closer to the boundary of stability domain, the autocovariance matrix gets singular and hence the condition number of the auto-covariance matrix can get arbitrarily large. Proposition 1, therefore, can be interpreted as if the underlying process gets closer to the boundary of the stability domain the causal and statistical errors can diverge.

For intuition, let us revisit our motivating example from the introduction with strongly correlated observations in an $AR(p)$ process. Let, without loss of generality $p = q$. Introducing the vectors $a := (a_1, a_2, \ldots, a_p)$ and $\hat{a} := (\hat{a}_1, \hat{a}_2, \ldots, \hat{a}_p)$ and the covariance matrix $\Sigma_p = \Sigma_{\max\{p,q\}}$. Then the quotient between causal and statistical error for predicting one time step ahead i.e. ($\omega = 1$) reads:

$$\frac{\mathcal{G}_{do_\omega}}{\mathcal{S}_\omega} = \frac{(\hat{a} - a)^T(\hat{a} - a) + \sigma_\epsilon^2}{(\hat{a} - a)^T\Sigma_p(\hat{a} - a) + \sigma_\epsilon^2}, \qquad (10)$$

Where we have assumed $X_t$ to have unit variance without loss of generality. The quotient is maximized if $(\hat{a} - a)$ is a multiple of the eigenvector to the smallest eigenvalue of $\Sigma_p$. This aligns with the intuition that causal loss diverges when the auto-covariance matrix gets singular. Moreover, we see that the vector $(\hat{a} - a)$ can be large with little observable effect when it mainly consists of eigenvectors with small eigenvalues of $\Sigma_p$. In the extreme case, if the minimum eigenvalue of the autocovariance matrix is 0, it is possible to arbitrarily deviate from the true model parameters along the direction of the corresponding eigenvector which can significantly affect the causal error without affecting the statistical error at all. For an $AR(2)$ process, for instance, we obtain $\Sigma_p = \begin{pmatrix} 1 & a_1/(1-a_2) \\ a_1/(1-a_2) & 1 \end{pmatrix}$, which becomes singular for $a_1 = \pm(1 - a_2)$ which indeed is the boundary of the stability domain (see for example, Lütkepohl (2009)). This is the limit in Section 1 where $X_t = \pm X_{t-1}$. The eigenvector for eigenvalue 0 reads $(1, \mp 1)$. Accordingly, the quotient (10) diverges when $\hat{a}$ differs from $a$ by $(1, \mp 1)$.

This further highlights that even for simple classes of forecasting models and with simplifying assumptions such as causal sufficiency, causal risks may even diverge from statistical risks. To show this formally, by means of Lemma 2, we can derive an explicit upper bound on the condition number of the autocovariance matrix $\kappa(\Sigma_{\max\{p,q\}})$ for AR(p) models and arrive at Corollary 2.

**Corollary 2 (Stability Controls Causal Generalization (AR)).** *Consider an AR(q) process, such that eigenvalues of its companion matrix satisfy $|\lambda| < \delta < 1$. For any AR(q)*

*model $f$,*

$$|\mathcal{G}_{\omega,i} - \mathcal{S}_\omega| \leq K_p \mathcal{S}_\omega(f)\nu(1 + \delta)^{2\nu}/(1 - \delta^2), \qquad (11)$$

*where $K_p$ is some finite constant that depends on the order $p$ of the underlying process.*

The bound in Corollary 2 is elegant due to its simplicity and generality. However, the cost of generality of the bound that relies only on the stability parameter is clearly that it cannot explain the variations in behavior exhibited by individual processes with the same stability parameter. For instance, consider an AR(2) model with parameters $a_1$ and $a_2$ with $a_2 \approx 0$ so that it is essentially an AR(1) model. Then, it is easy to verify that $\lambda_2 \approx 0$. The combinatorial definition of the Schur polynomials (Macdonald 1998) allows us to express the coefficients as follows: $A_{11}^\omega = \sum_{i=0}^\omega \lambda_1^{\omega-i}\lambda_2^i$, $A_{12}^\omega = \sum_{i=1}^{\omega-1} \lambda_1^{\omega-i}\lambda_2^i$. Combining this with Corollary 1, it is easy to see that if the estimated model is also close to AR(1), then the coefficients $A_{12}^\omega$ and $\widehat{A}_{12}^\omega$ and hence the difference in statistical and causal errors is close to 0. The bound in (11) which relies on the stability parameter does not capture this. For tighter bounds that utilize additional information about the spectrum of the companion matrix, we can exploit the connection to Schur polynomials to arrive at the following bound.

$$|\mathcal{G}_{\omega,i} - \mathcal{S}_\omega| \leq K_{p,q} \max\left\{\delta, \widehat{\delta}\right\}^\omega \sum_{k=2}^\nu \left(S_{\omega k}^\lambda - S_{\omega k}^{\hat{\lambda}}\right)\gamma_{k-1},$$

where $K_{p,q}$ is a constant that depends on $p, q, \delta$ and $\widehat{\delta}$ are the stability parameters of the true and estimated processes respectively and $\lambda$ and $\widehat{\lambda}$ denote the set of eigenvalues of $A$ and $\widehat{A}$ respectively.

# 4 SIMULATIONS

To verify the practical behavior of causal and statistical risks, we provide some simple simulations to study the errors of different estimators under AR processes. For each presented plot, we draw parameters for 10,000 stationary $AR(p)$ processes using rejection sampling. We draw the coefficients of each process independently and uniformly from $[-2, 2]$ and reject sets of parameters that yield a non-stationary process. For each process, we draw a training sample with 100 timesteps and a test sample with 1000 timesteps. For all figures in the main paper we set $\omega = 1$. To estimate the coefficients we use Ordinary Least Squares (OLS). In Appendix 5 we provide additional plots with hidden confounder, as well as varying order, sample size, $\omega$ and other estimators: Ridge, Lasso, and Elastic Net regressors. OLS minimizes the empirical statistical error, that is, $\sum_{y_i,\hat{y}_i}(y_i - \hat{y}_i)^2$, where $\hat{y}_i$ denotes the model prediction with estimated parameters $\hat{a}$.

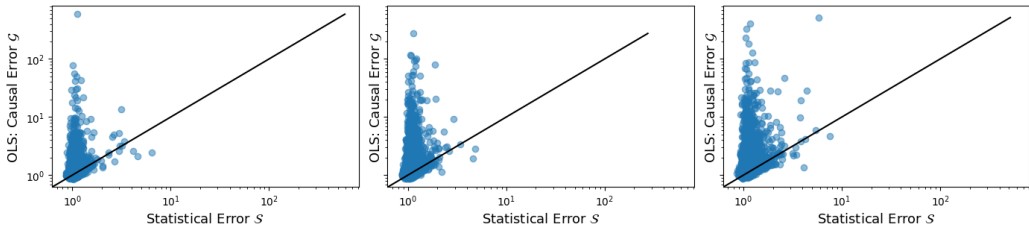

Figure 2: The causal error $\mathcal{G}$ versus the statistical error $\mathcal{S}$ for AR($p$) processes with $p = 3, 5, 7$.

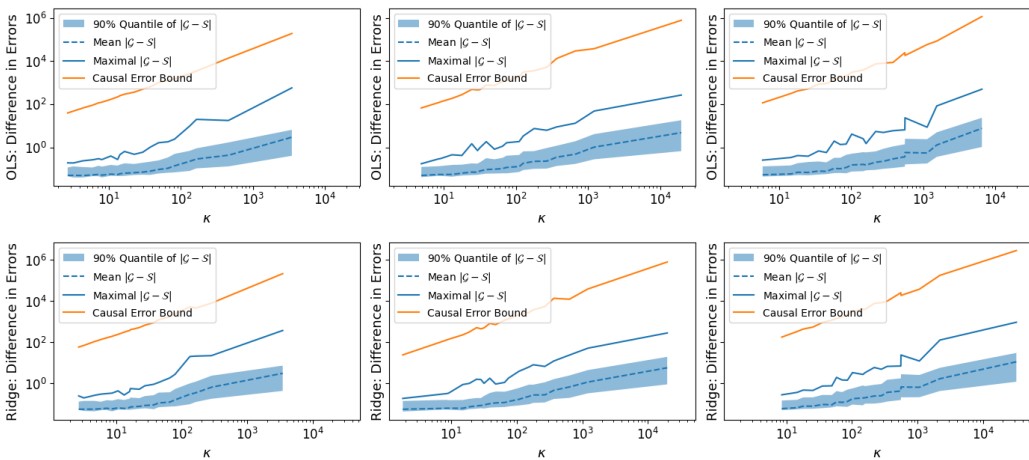

Figure 3: The maximal difference between statistical error $\mathcal{S}$ and causal error $\mathcal{G}$ as well as an estimate for the generalization bound in Theorem 1 for increasing condition number $\kappa$ for process orders $p = 3, 5, 7$ (from left to right). The maximum is taken over 500 datasets with the closest $\kappa$. Our theoretical bounds (orange) closely match the empirical evaluations up to constant factors (blue).

In line with our theoretical results, we find that even for simple scalar AR processes of small orders, the causal error of the estimators is often several times larger than the statistical error (see Figure 2). In Figure 3 we sorted the randomly drawn datasets by their autocorrelation (measured by the condition number $\kappa$ of the autocorrelation matrix) and split the sorted list into buckets of 500 dataset. For each we calculated the maximum, mean and 90% quantile of the difference in causal and statistical error for the OLS and Ridge estimators. The plots corresponding to the other estimators are provided in Appendix 5 We can see that upto constant factors, our theoretical finite sample causal generalization bound matches the difference in causal and statistical risks observed empirically.

## 5 EXPERIMENTS ON REAL DATA

**Data.** We conduct experiments on three different datasets: m4 hourly (Makridakis et al. 2018), electricity (Dua et al. 2017), and traffic (Dua et al. 2017). The m4 hourly dataset includes timeseries from a diverse set of sources. The m4 dataset has a hourly frequency and a prediction length of 48. The traffic dataset records the occupancy rates of car lanes on freeways in the San Francisco Bay Area and the

electricity dataset records the electricity consumption of 370 customers hourly. To create an interventional distribution without a generative model, for each time series we replace the last time step prior to the evaluation window by sampling at random either from all time-series at that time step (referred to as *across-ts*) or from previous values of the same time series (referred to as *within-ts*).

**Models.** We include three popular deep neural network architectures in our evaluation. DeepAR consists of an RNN that takes the previous time steps as inputs and predicts the parameters of an auto-regressive model (Gasthaus et al. 2019). Wavenet is a hierarchical CNN developed for speech-to-text (Oord et al. 2016). Transformer is an attention-based deep neural network widely applied to NLP tasks including translation (Vaswani et al. 2017). For all these models we use AutoGluonTS's default hyperparameters.

The experiments were conducted using GluonTS (Alexandrov et al. 2019) with default hyperparameters on instances with 4 virtual CPUs and a 2.9 GHz processor. The code for reproducing all the experiments can be found at `https://github.com/amazon-research/causal-forecasting`

**Metrics.** For the observational distribution, we compute

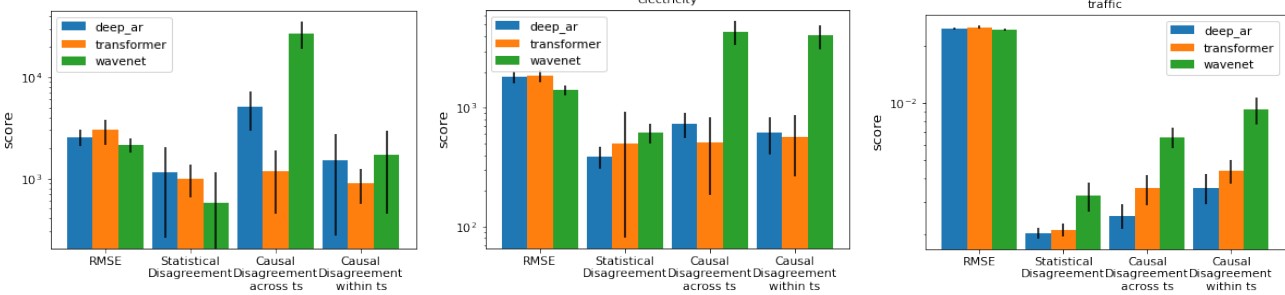

Figure 4: Results of the evaluation of three different deep neural network architectures on the m4hourly, electricity, and traffic datasets. The "RMSE" is computed comparing prediction on the observational data against the ground truth. The disagreement from Def. 5.1 compares the root-mean-square deviation between the predictions of two models of the same architecture on the observational data ("Statistical Disagreement") and interventional distributions ("Causal Disagreement Across TS" sampling interventions from all of time-series and "Causal Disagreement Within TS" sampling interventions from prior points within the time series). The results are averaged over 5 runs of training and evaluation and include standard deviation in black.

the root-mean-square error (RMSE) comparing average prediction for each time point with the ground truth in the evaluation set. For the interventional distribution we are lacking ground truth. Therefore, we train two separate models and compute their disagreement.

**Definition 5.1.** The disagreement is the average root-mean-square deviation of the mean forecasts of two models. The average is taken over a set of time-series. If the time-series come from the original dataset, we call it the statistical disagreement. If they come from one of the interventional datasets, we call it causal disagreement and specify the type of intervention as across time-series or within time-series.

This disagreement is a measure of uncertainty introduced by the randomness in the training and evaluation procedure. Here, however, we use it to approach the causal risk, that we cannot compute directly. If the disagreement is high on the interventional distribution at least one of the models must have a high causal risk. For comparison, we also included this disagreement measure for examples from the observational distribution. Finally, to explore the relationship between causal forecasting error and uncertainty, we also compute the width of the 80% prediction interval for both the observational and interventional distribution.

**Definition 5.2.** The 80% prediction width of a forecasting model is the absolute distance between the 0.9 quantile and 0.1 quantile of the forecast distribution. It is averaged over a set of time-series that can come from the observational or the interventional distritibutions.

**Limitations.** The dataset and models have clear shortcomings. Likely, the dataset is not causally sufficient. Also, we did not tune the models. Moreover, we are lacking samples from the marginal distribution for the interventions and groundtruth on what happens under these interventions.

| Model | observ. | across-ts interv. | within-ts interv. |
|---|---|---|---|
| DeepAR | $940.0 \pm 126.2$ | $1329.2 \pm 187.5$ | $953.1 \pm 124.2$ |
| wavenet | $1253.9 \pm 96.6$ | $3444.7 \pm 649.4$ | $1612.7 \pm 257.7$ |
| transformer | $1259.3 \pm 139.3$ | $1355.1 \pm 129.6$ | $1255.7 \pm 139.3$ |

Table 1: 80% prediction width for the m4 dataset, see Def. 5.2, for observational and interventional forecasts. Averaged over 5 runs with std.

Nevertheless, we hope to get a sense for how popular deep learning networks can behave on real data for relevant prediction tasks under interventions.

**Results.** Figure 4 shows the results of the metrics when we evaluate the models on the datsets for both observation and interventional distributions. We see that the causal disagreement between two models of the same architecture and hyper-parameters can be much higher than their disagreement on the observational distribution. While there are only smaller differences in the statistical risk between the model architectures, their causal disagreement differs more. Overall, the the causal disagreement can be high, which implies high causal risk, but it varies across datasets and model architectures. Wavenet's disagreement is an order of magnitude larger when sampling interventions from other time-series. For transformer models their interventional disagreement is close to the observational one.

**Uncertainty.**

When we compare the width of the 80% interval of predictions in Table 1 (m4 dataset) and Table 2 (electricity and traffic datasets) we see that this uncertainty measure is higher for the interventional distribution compared to the observational one. Moreover, directionally it relates to the causal disagreement across models. Unlike the disagreement that requires a second model to be trained, this uncertainty measure is readily available from the predicted forecasts.

| Dataset | electricity | | | traffic | | |
|---|---|---|---|---|---|---|
| Model | observ. | across-ts interv. | within-ts interv. | observ. | across-ts interv. | within-ts interv. |
| DeepAR | $381.550 \pm 21.647$ | $449.781 \pm 27.536$ | $375.632 \pm 20.851$ | $0.0282 \pm 0.0015$ | $0.0288 \pm 0.0017$ | $0.0294 \pm 0.0018$ |
| wavenet | $470.691 \pm 15.886$ | $799.307 \pm 65.722$ | $588.469 \pm 39.911$ | $0.0246 \pm 0.0003$ | $0.0279 \pm 0.0003$ | $0.0299 \pm 0.0003$ |
| transformer | $413.174 \pm 31.243$ | $575.946 \pm 35.456$ | $407.372 \pm 29.073$ | $0.0282 \pm 0.0023$ | $0.0312 \pm 0.0031$ | $0.0328 \pm 0.0033$ |

Table 2: 80% prediction width for observational and interventional forecasts on electricity and traffic datasets. Averaged over 5 runs with std.

The causal disagreement can be high for some models which implies a high causal risk. This cautions against the use of statistical deep learning models to forecast what will happen under interventions. The difference we observe in causal disagreement across models motivates further development of specific model architectures suitable for causal forecasting. For existing models, the uncertainty measure considering the width of the prediction interval can be an indicator for causal risk.

# 6 RELATED WORK

Our work intersects with domain adaption, RL, and treatment effect estimation, reviewed separately below.

**Domain Adaptation.** The literature closest to our setting is that of learning theory for domain adaptation, in particular, for covariate shift. Theoretical analysis of domain adaptation when labelled samples from the source distribution and unlabelled samples from the target distribution are generated i.i.d was initiated by Ben-David et al. (2007), who provided VC bounds for binary classification under covariate shifts based on a *discrepancy measure* $d_{\mathcal{F}}$ between source and target distributions that depends on the hypothesis class $\mathcal{F}$ and is estimable from finite samples. Mansour et al. (2009) extended the work to the context of regression in the i.i.d setting by adapting the discrepancy measure for more general loss functions and by providing tighter, data-dependent Rademacher bounds. Despite the i.i.d assumption, the results in Mansour et al. (2009) are perhaps the most relevant to our setting. We can utilize one of the main results from Mansour et al. (2009, Theorem 8) which does not rely on the i.i.d assumption to arrive at the following population-level bound for our setting: $|\mathcal{G}_{\omega,i}(f, f^*) - \mathcal{S}_{\omega}(f, f^*)| \leq \sup_{f,f' \in \mathcal{F}} |\mathcal{G}_{\omega,i}(f, f') - \mathcal{S}_{\omega}(f, f')|$. These bounds are non-informative in our context since they do not incorporate structural knowledge of the class of interventional distributions under a VAR model.

**Estimation of Treatment Effects.** A related problem is that of estimating treatment effects in the potential outcomes framework (Hill et al. 2006; Shi et al. 2019), where the goal is to estimate the effects of binary-valued treatments from observational data under a multivariate confounding model. Our setting is more general in that variables in the multivariate process can take a continuum of interventions and play a multiplicity of roles — each variable plays the role of

treatment, confounder, and the target variable. Of particular relevance is the work of Shalit et al. (2017) and Johansson et al. (2020), who prove generalization error bounds on estimating individual-level treatment effects in terms of standard generalization error and a distance measure between the treated and control distributions. This result is similar to domain adaptation bounds in Ben-David et al. (2007) and Mansour et al. (2009) and may be interpreted as causal learning theory in the sense of our paper.

**Reinforcement Learning.** The ratio of observational versus interventional densities in our setting play a similar role as the state density ratio in off-policy evaluation in reinforcement learning(RL) (Bennett et al. 2021). In RL, however, the clear separation between the state of actions and the state space acted on admits techniques that we do not see for our problem, e.g., deconfounding (Hatt et al. 2021), or learning representations of the history that are independent of the actions (Bica et al. 2020), which overcomes the problem of high inverse probability weightings (Lim et al. 2018).

# 7 DISCUSSION AND CONCLUSION

Our work highlights that even for very simple models and even under simplifying assumptions such as causal sufficiency, causal and statistical errors can diverge. It emphasizes the need for providing guarantees for causal generalization in a similar vein as providing guarantees for statistical learning. To this end, we initiate a first analysis in this direction by introducing a framework for causal learning theory for forecasting and providing conditions under which one can guarantee generalization in the causal sense for the class of VAR models. We hope that this work inspires more theoretical work that allows certifying the validity of the causally interpreting forecasting models.

Our theoretical as well as empirical results challenge the causal interpretation of forecasting models used in practice which are typically far more complex. Our experiments show that causal disagreement can be high for some models which implies a high causal risk. This cautions against the use of statistical deep learning models for causal forecasting. The difference we observe in causal disagreement across models motivates further development of specific model architectures suitable for causal forecasting. For existing models, the uncertainty measure considering the width of the prediction interval can be an indicator for causal risk.

# 8 ACKNOWLEDGMENTS

This work has been supported by the German Research Foundation (Research Training Group GRK 2428) and the Baden-Wurttemberg Stiftung (Eliteprogram for Postdocs project "Clustering large ¨ evolving networks"). The authors thank the International Max Planck Research School for Intelligent Systems (IMPRS-IS) for supporting Leena Chennuru Vankadara.

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
