# OpenReview forum: "Causal Forecasting: Generalization Bounds for Autoregressive Models"
_auai.org/UAI/2022/Conference — UAI 2022 Poster_

### Official Review · Reviewer_gaMX · 2022-03-30

**Q2(1) Originality/Novelty:** 3
**Q2(2) Significance/Impact:** 3
**Q2(3) Correctness/Technical Quality:** 3
**Q2(6) Clarity Of Writing:** 3
**Q6 Overall Score:** 6
**Q8 Confidence In Your Score:** 3

**Q1 Summary And Contributions:**

This paper investigate the causal generalization of VAR model. It build some conclusion on the gap between statistic risk and causal risk. From the theoretical analysis, it can be seen that the conditional number of autocovariance matrix have   great influence on the divergence of statistic risk and causal risk. Further interpretation based on the theory also make sense. The  experimental results show that the theoretical analysis coincide the empirical result.

**Q2 Assessment Of The Paper:**

More detailed information regarding each of these aspects is given below:

**Q2(4) Quality Of Experiments (Optional):**

3: Good: The experimental evaluation is adequate, and the results convincingly support the main claims.

**Q2(5) Reproducibility:**

2: Fair: Key resources (e.g., proofs, code, data) are unavailable but key details (e.g., proof sketches, experimental setup) are sufficiently well-described for an expert to confidently reproduce the main results.

**Q3 Main Strengths:**

1. The causal forecasting is an important and interesting problem for many applications. It also has strong relationship with other fields.
2. The paper gives thorough and meticulous theoretical results, which gives great inspiration to readers. The extended interpretation also make senses. I believe the results are correct.
3. The experimental results are sufficient and firmly supported the theoretical analyisis.

**Q4 Main Weakness:**

The presentation of the paper need improvement. The metrics in experiments of the real world dataset need more description.
The reproducibility of the paper is kind of weak. The hyper-parameters of different models is not revealed. And I hope the authors could release the code if the paper would be accepted in the future.


**Q5 Detailed Comments To The Authors:**

How to compute the metric "score"  and the degree of disagreement is confusing. And the results in Table 1 is also not clear. Does the number mean average prediction output? If it does, how to calculate the average prediction?

**Q7 Justification For Your Score:**

I think it is a solid paper with sufficient theoretical results. However, the paper is not reader-friendly enough. Overall, I suggest a postive score.

**Q9 Complying With Reviewing Instructions:**

1: Yes.

---

### Official Review · Reviewer_RQBf · 2022-04-08

**Q2(1) Originality/Novelty:** 4
**Q2(2) Significance/Impact:** 4
**Q2(3) Correctness/Technical Quality:** 3
**Q2(6) Clarity Of Writing:** 4
**Q6 Overall Score:** 7
**Q8 Confidence In Your Score:** 3

**Q1 Summary And Contributions:**

In this paper, the authors consider the problem of causal generalization in which the generalization from the observation to the interventional distributions is considered. Hence, they propose the theory that bridges the statistical and causal risk. First, the authors introduce the definition of causal errors, then they respectively provide the generalization bounds for VAR and AR.

**Q2 Assessment Of The Paper:**

More detailed information regarding each of these aspects is given below:

**Q2(4) Quality Of Experiments (Optional):**

3: Good: The experimental evaluation is adequate, and the results convincingly support the main claims.

**Q2(5) Reproducibility:**

3: Good: Key resources (e.g., proofs, code, data) are available and key details (e.g., proofs, experimental setup) are sufficiently well-described for competent researchers to confidently reproduce the main results.

**Q3 Main Strengths:**

1.	The authors consider the problem of causal generalization, which is important in many fields like domain adaptation and reinforcement learning.
2.	The authors provide the first theoretical guarantees for the causal generalization in the time-series setting.
3.	The experiment results valuate the proposed theory and are sound.

**Q4 Main Weakness:**

1.	Some contributions are unclear. For example, it is confusing if Lemma 1 is a contribution.
2.	There are some typos, for example, （P_{y|x=x*} should be P(y|x=x*).


**Q5 Detailed Comments To The Authors:**

1.	It seems that the authors only consider the linear case, it is suggested that the authors should consider the nonlinear case.

**Q7 Justification For Your Score:**

I also focus on how to bridge the causality and the statistical machine learning from the theoretical perspectives. I think this is an interesting and important work that inspires us to consider the generalization by measuring the difference between the causal graph.

**Q9 Complying With Reviewing Instructions:**

1: Yes.

---

### Official Review · Reviewer_tG5w · 2022-04-12

**Q2(1) Originality/Novelty:** 3
**Q2(2) Significance/Impact:** 2
**Q2(3) Correctness/Technical Quality:** 3
**Q2(6) Clarity Of Writing:** 3
**Q6 Overall Score:** 6
**Q8 Confidence In Your Score:** 4

**Q1 Summary And Contributions:**

This paper studies the problem of causal generalization that is understanding the difference between interventional and observational distributions. The results are introducing a framework for causal learning which helps them to characterize the difference between the statistical and causal risks. They also conducted several experiments to verify their theoretical findings.

**Q2 Assessment Of The Paper:**

More detailed information regarding each of these aspects is given below:

**Q2(4) Quality Of Experiments (Optional):**

3: Good: The experimental evaluation is adequate, and the results convincingly support the main claims.

**Q2(5) Reproducibility:**

2: Fair: Key resources (e.g., proofs, code, data) are unavailable but key details (e.g., proof sketches, experimental setup) are sufficiently well-described for an expert to confidently reproduce the main results.

**Q3 Main Strengths:**

The paper is well-written. The discussions in Section 3 are helpful to make the results more readable.  It studies an interesting research problem.

The contribution is novel. Although it only considers a simple class of VAR models for detailed analysis but building a framework for understanding the causal error and characterizing it for VAR models can be used by other researcher for further studies and perhaps extend the theoretical results to other function classes.

The theoretical results are interesting as they confirm our intuitions about the relations between correlations, post-interventional distributions, and observational distributions.




**Q4 Main Weakness:**

My main concern about the motivation of this work. In causal inference literature and more specifically in causal identification, estimating a post interventional or a counterfactual distribution from observational distribution (given the causal graph) is quite important and up to some extend well understood.








**Q5 Detailed Comments To The Authors:**

It is understood and expected that an interventional distribution e.g., P(y|do(x)) is possibly different from its conditional counterpart, P(y|x). Thus, it is not expected that the statistical forecast error, i.e., Eq (1) and the causal forecast error EQ (2) are close in general.
Understanding conditions under which these two objects are close (the main contribution of this work) is, I believe, valuable but not the goal. On the other hand, estimating the causal forecast, e.g., estimating treatment effect, from observational distributions has more applications.
For example, given the result of Corollary 1, we are able to say whether causal and statistical forecast errors are close or not based on model parameters of a VAR. However, if we had all these parameters, estimating the causal forecast would have been a more interesting problem.

Given the experimental results, it seems the gap between the theoretical bounds and the estimated error is a large constant.


Minor comments:

In the notation section: y_t^n should be x_t^n.
Above equation (5): x_{t-i} should be x_{t-1}.
Lemma 2: definition of set K is unclear.
Top of page 5: A^k should be A^\omega.

**Q7 Justification For Your Score:**

Given the proposed framework for studying the causal and statistical forecast errors and also the theoretical results for VAR models, I believe this work has interesting and novel results. However, there is concern about the main objective which requires more motivations.

**Q9 Complying With Reviewing Instructions:**

1: Yes.

---

### Official Review · Reviewer_qSiu · 2022-04-13

**Q2(1) Originality/Novelty:** 3
**Q2(2) Significance/Impact:** 3
**Q2(3) Correctness/Technical Quality:** 2
**Q2(6) Clarity Of Writing:** 3
**Q6 Overall Score:** 6
**Q8 Confidence In Your Score:** 3

**Q1 Summary And Contributions:**

The authors show that even with simplifying causal assumptions, causal and statistical errors can diverge. They investigate how forecasting with and without interventional data changes the results and derive theoretical bounds to that end.

**Q2 Assessment Of The Paper:**

More detailed information regarding each of these aspects is given below:

**Q2(4) Quality Of Experiments (Optional):**

3: Good: The experimental evaluation is adequate, and the results convincingly support the main claims.

**Q2(5) Reproducibility:**

2: Fair: Key resources (e.g., proofs, code, data) are unavailable but key details (e.g., proof sketches, experimental setup) are sufficiently well-described for an expert to confidently reproduce the main results.

**Q3 Main Strengths:**

- Technically solid work
- Good justification
- Good structure of paper

**Q4 Main Weakness:**

- It is hard to follow for non-experts
- More work should be put in to make it more accessible (and hence more usable) for 'outsiders'
- I think it would help the paper if it were dotted with specific examples which other investigators may be more familiar with and how these results impact these settings -- anything from Pearl's book (2009) will do the job as those results are sufficiently mature and (mostly) known by investigators
- Parts of the paper are needlessly verbose (e.g. abstract and contributions)

**Q5 Detailed Comments To The Authors:**

# Abstract

- General point: you're pushing the boundary on what is an acceptable length for an abstract. Make it shorter. They are meant to summarise your paper. Nothing more.

# Introduction

- You say "While both models are similar in their statistical predictions, they differ substantially in their causal predictions" -- I find it strange that you have not yet referenced do-calculus, which deals with the interaction of observational and interventional distributions and how to get to latter from the former. I also do not understand then why you are not using the associated vocabulary like _interventional_ distribution. Further it not clear what you are trying to get at when you say "causal and statistical predictability can differ significantly". Well yes, they are fundamentally different statistical expressions: $p(y \mid x)$ vs $p(y \mid \text{do}(x))$. Given this machinery and its maturity in the field, why are you not using it to describe your problem? The writing is thus not clear to me.

# CAUSAL LEARNING THEORY FOR FORECASTING

- Your second paragraph on page three, bears resemblance to Granger causality. It may be worthwhile to investigate the links and cite the relevant literature. This is not "real" causality of course but one time-series is used to predict another time-series or point.
- Style: consider using widehat instead of hat in your symbols e.g. $\widehat{S}$
- Writing: add full-stop after sentence 'that achieves the optimal causal risk $\mathcal{G}$ ...'
- Style: consider using $\mathsf{T}$ for the tranpose
- General question: how do different types of stationarity (strict, first-order, second-order and so on) affect your results?

# CAUSAL GENERALIZATION FOR VAR

- A small example would help after equation (8) to aid understanding. A simple histogram would be sufficient.
- Notation: you're multiplication symbol in lemma two is wrong, a backslash is missing in $K= ...$
- Writing: when referencing a body of work you use citep not citet w.r.t. Chaugule et al. 2019
- Is there a reference for the Weyl formulation of Schur polynomials?
- What do you mean by: "While this is not the most interpretable definition per see"
- Readability: please do not put matrices inline, use the equation environment for $\sigma_p$ - it is very difficult to read otherwise

# SIMULATIONS

- Will the code be made available?
- Please explain this "reject sets of parameters that yield a non-stationary process" I don't follow, shouldn't your draw only produce parameters which result in stationary processes?

# EXPERIMENTS ON REAL DATA

- What parameterization did you use for your models?
- Figure 2 could use a lot more explanation. Why does it look the way it does? What is the black line? No figure has a legend. You need to explain what we are looking at.

**Q7 Justification For Your Score:**

I am not an expert in this field and have not checked all the technical details. Given my understanding and the outstanding questions that I have, I believe this work is a good contribution to the existing literature, which will spur new directions in causal thinking and subsequent work in this direction.

**Q9 Complying With Reviewing Instructions:**

1: Yes.

---

### Decision · Program_Chairs · 2022-05-15

**Decision:**

Accept (Poster)

**Comment:**

Meta Review: Thank you for your submission to UAI and author response to reviews.

This paper investigates the causal generalizability of VAR models, and the gap between statistical risk and causal risk.  The work shows that even simple models can show divergent statistical and causal errors.  It presents a theoretical analysis that shows, under certain conditions, can guarantee causal generalizability for forecasting scenarios under interventions.

The reviewers appreciated the focus of this paper, recognizing its importance for many applications and its connections to related fields such as reinforcement learning.  Reviewers highlighted a number of questions and feedback, including suggestions for making the work more accessible for non-experts, questions about the motivation, and reproducibility of the paper.   We appreciate the authors responses, and their responsive updates to the paper.